# Gallium Nitride (GaN) Nanostructures and Their Gas Sensing Properties: A Review

**DOI:** 10.3390/s20143889

**Published:** 2020-07-13

**Authors:** Md Ashfaque Hossain Khan, Mulpuri V. Rao

**Affiliations:** Department of Electrical and Computer Engineering, George Mason University, Fairfax, VA 22030, USA; rmulpuri@gmu.edu

**Keywords:** gallium nitride (GaN), nanostructure, gas sensing, sensitivity, response/recovery time, density-functional theory (DFT), internet of things (IoT), machine learning (ML)

## Abstract

In the last two decades, GaN nanostructures of various forms like nanowires (NWs), nanotubes (NTs), nanofibers (NFs), nanoparticles (NPs) and nanonetworks (NNs) have been reported for gas sensing applications. In this paper, we have reviewed our group’s work and the works published by other groups on the advances in GaN nanostructures-based sensors for detection of gases such as hydrogen (H_2_), alcohols (R-OH), methane (CH_4_), benzene and its derivatives, nitric oxide (NO), nitrogen dioxide (NO_2_), sulfur-dioxide (SO_2_), ammonia (NH_3_), hydrogen sulfide (H_2_S) and carbon dioxide (CO_2_). The important sensing performance parameters like limit of detection, response/recovery time and operating temperature for different type of sensors have been summarized and tabulated to provide a thorough performance comparison. A novel metric, the product of response time and limit of detection, has been established, to quantify and compare the overall sensing performance of GaN nanostructure-based devices reported so far. According to this metric, it was found that the InGaN/GaN NW-based sensor exhibits superior overall sensing performance for H_2_ gas sensing, whereas the GaN/(TiO_2_–Pt) nanowire-nanoclusters (NWNCs)-based sensor is better for ethanol sensing. The GaN/TiO_2_ NWNC-based sensor is also well suited for TNT sensing. This paper has also reviewed density-functional theory (DFT)-based first principle studies on the interaction between gas molecules and GaN. The implementation of machine learning algorithms on GaN nanostructured sensors and sensor array has been analyzed as well. Finally, gas sensing mechanism on GaN nanostructure-based sensors at room temperature has been discussed.

## 1. Introduction

Despite having many advantages, carbon nanotubes (NTs) still suffer from the uncontrollability of selective growth of semiconducting and metallic NTs [1]. Also, the bandgap of carbon NTs cannot be controlled, hence their chemical reactivity cannot be influenced strongly [2]. Semiconducting nanowires made of compound semiconductors, possess higher flexibility as their bandgaps and doping characteristics can be tuned by manipulating the alloy composition and impurity concentrations, respectively [3]. Compound semiconductors are mainly prepared from the elements in groups II to VI of the periodic table. Several binary and ternary compound semiconductor-based nanostructure devices had been demonstrated earlier [4]. In case of group III–V compounds, each group III atom is attached to four group V atoms to achieve an octet in the valence band. The valence charge from a group V atom moves toward a group III atom and thus promotes partial ionic bonding to the crystal [5].

Bulk GaN is one of the most explored group III-nitride semiconductor materials, which has been employed in various applications such as optoelectronic devices [6], electronic devices [7], biosensors [8], chemical sensors [9,10,11] and so on. Since GaN has a direct and wide band gap of 3.4 eV at room temperature, it is quite robust. Other group III elements like Al and In can be alloyed with Ga to tune the bandgap of III-nitrides from 0.8 eV to 6 eV [12,13,14]. Moreover, it possesses high electron mobility, high heat capacity and high breakdown voltage, which are all useful properties for reliable sensing of gas/chemical molecules [15]. 

Internet of Things (IoT) applications require ultra-low power, mini-sized chemical sensors, which are easily integrated into electronic circuits for remote air quality monitoring in automated systems [16,17,18]. Nanostructures are suitable candidates for this type of sensing application. Having a large surface-to-volume ratio, nanostructures such as nanowires, nanorods, nanotubes, nanoparticles and nanobelts favor adsorption of gas molecules on the sensor and thus increase the sensitivity of the device [19,20,21]. The larger interaction between the gas analytes and the sensing surface allows nanostructures to be employed for high performance gas sensing as opposed to their bulk/microstructure counterparts. Although commercially available metal-oxide nanostructure-based gas sensors show high sensitivity and low detection limits [22], they suffer from issues such as poor analyte selectivity, high operating temperature and unstable performance in harsh environments [23]. GaN nanostructures offer stable operation under various radiation and in space conditions. They operate at room-temperature and can also tolerate large variations of temperature and humidity as compared to metal-oxides [24]. Thus, GaN nanostructure sensors have the potential to take over a significant share of the gas sensing market.

In the past several years, efforts have been made on GaN nanostructures-based devices to detect various chemical analytes. In this work, we have reviewed our group’s work and the works published by other groups on the advances in GaN nanostructures-based sensors for detection of gases such as hydrogen (H_2_), alcohols (R-OH), methane (CH_4_), benzene and its derivatives, nitric oxide (NO), nitrogen dioxide (NO_2_), sulfur-dioxide (SO_2_), ammonia (NH_3_), hydrogen sulfide (H_2_S) and carbon dioxide (CO_2_). The sensing performance parameters like limit of detection, response/recovery time and operating temperature for different types of sensors and structures have been summarized and tabulated to perform the comparative study. A novel metric, the product of response time and limit of detection, has been calculated for each sensor in order to compare the overall sensing performance. Then, DFT studies on molecular models of gas molecules and GaN have been reviewed. Next, photo-assisted gas sensing and machine learning implementation have been discussed. Furthermore, gas sensing mechanisms of the GaN sensors have been discussed to understand the basic interaction between sensor surface and gas molecules. 

## 2. GaN Nanostructures-Based Gas Sensors

### 2.1. GaN Nanostructures-Based Hydrogen Sensors

Hydrogen gas is highly inflammable when its concentration exceeds 4% in air [25]. Detection of H_2_ leakage from gas pipes and storage systems is thus very important to avoid explosions [26]. GaN NW networks decorated with Ga_2_Pd_5_ nanodots have been demonstrated as a high performance H_2_ sensor [27]. The Ga_2_Pd_5_ nanodot functionalization enhanced the response by more than 50-fold compared to that of bare GaN NWs due to two main reasons: chemical sensitization and electronic sensitization mechanisms [28]. The hydrogen spillover effect of Ga_2_Pd_5_ played an important role in this enhancement, where atomic hydrogen is generated through the catalytic dissociation of H_2_ molecules. The fabricated device was able to detect as low as 100 ppm of H_2_ gas at room temperature. However, the recovery process of the sensor was quite slow (800 s).

Aluri et al. [29] fabricated TiO_2_-Pt nanocluster-coated GaN NW sensor devices. Figure 1A shows a HRTEM image of TiO_2_ sputtered GaN NW after Pt deposition, where titania (TiO_2_) particles exhibits 0.35 nm fringes corresponding to (101) lattice spacing of the anatase polymorph. The black arrows indicate an amorphized surface film having a thickness of 2–5 nm. The device was able to detect as low as 1 ppm of H_2_ gas at room temperature with a short response-recovery time (60 s/45 s) under UV illumination. Figure 1B shows the variation of sensor current of a GaN/(TiO_2_–Pt) device for different concentrations of H_2_ in nitrogen. The sensing behavior was mainly attributed to the work-function change of Pt NCs due to hydrogen adsorption [30]. 

Abdullah et al. [31] reported that GaN adopts various morphologies under various NH_3_ gas flow rates at constant temperature (1000 °C) during its chemical vapor deposition (CVD) growth process. GaN was grown as thin films, nanowires and microstructures, respectively, with increasing NH_3_ flow rate. The nanowires-based sensor exhibited a response of 127% for 100 ppm H_2_ gas at room temperature, which is quite high compared to the corresponding microstructures-based sensor. The high surface area of the GaN NWs facilitated this sensor performance enhancement. Paul et al. [32] developed Pt-film decorated InGaN/GaN NWs for H_2_ detection. The sensor was able to detect as low as 200 ppb of H_2_ at 80 °C with a fast response-recovery process (1 min/1 min). In another study, GaN NWs coated with Pt were employed for ppm level detection of H_2_ at room temperature [33]. It was found that response-recovery process of the sensor became faster with increasing operating temperature. Also, it was reported that the GaN nanowires co-decorated with Au and Pt nanoparticles show much stronger response to H_2_ gas than the Au or Pt monometal-decorated counterparts [34].

Beside nanowire-based sensors, other GaN nanostructures such as nanotube-, nanonetwork-, and nanoparticle-based H_2_ sensors have been explored as well. Sahoo et al. [35] synthesized wurtzite structured GaN nanotubes utilizing a quasi-vapor-solid process to detect H_2_ gas at room temperature. Pt nanoclusters were incorporated onto the nanotubes in order to enhance the sensitivity. The nanotube-based sensor was able to detect as low as 25 ppm of H_2_ with a small activation energy of 29.4 meV. In another work, a Pd coating was used to functionalize a network of GaN NWs for H_2_ sensing [36]. The as-grown GaN nanowire network is shown in Figure 1C. On exposure to different H_2_ concentrations at room temperature, the Pd-coated and uncoated GaN NW network sensors exhibited a clear contrast in response, as illustrated in Figure 1D. Pd has the capability to split hydrogen molecule into atoms, thus it contributes to the better sensor response.

In another study, a lightly Mg-doped porous GaN nanonetwork had been synthesized by a molecular beam epitaxy method [37]. Pt was deposited on a honeycomb nanonetwork by sputtering to form a network containing nano-Schottky contacts. The developed sensor was able to detect as low as 320 ppm of H_2_ gas at room temperature with a short response time (1 min). However, the recovery process of the sensor was quite slow (8 min). Furthermore, a GaN nanoparticles (NPs)-based thick film sensor successfully detected 50 ppm of H_2_ gas at room temperature with a fast recovery process (70 s) [28].

The GaN nanostructure sensor methods of synthesis and the corresponding sensing performance metrics, including limit of detection, response/recovery times and operating temperatures are summarized in Table 1. It provides a brief comparative performance outline among the different GaN nanostructure-based H_2_ sensors reported to date.

### 2.2. GaN Nanostructures-Based Alcohol Sensors

Because more than a certain amount of alcohol exposure is detrimental to health, the precise detection of ambient alcohol particles is necessary. For example, methanol can poison the human central nervous system with a median lethal dose of 100 mL [43]. Ji et al. incorporated GaN nanograins on a silicon nanoporous pillar array (Si-NPA) by the CVD technique for methanol detection [44]. The fabricated device was able to detect as low as 5 ppm of methanol with a fast response-recovery process due to a large surface area and numerous surface-active sites. The optimum temperature for the best sensor response was found to be 350 °C. 

Luo et al. [45] reported a comparative study on ethanol sensing performance between GaN nanofibers (NFs) and GaN nanoparticles (NPs). Because GaN-NFs provide porous structures, large specific surface areas and crosslinking properties, they show a better sensing response to ethanol [46]. The optimum working temperature found for both types of sensors was 320 °C. The synthesized sensors were capable of detecting 50 ppm of ethanol with a very fast response-recovery process (8 s/5 s). Moreover, the GaN-NFs-based sensor responded more quickly to ethanol than GaN-NPs-based sensor due to the fast diffusion capability of its porous nanostructures.

The alcohol sensing properties of GaN NWs grown on Si (1 1 1) substrates and functionalized with tin dioxide (SnO_2_) nanoparticles had been demonstrated by Bajpai et al. [47]. This chemiresistive sensor successfully detected alcohol vapors, including methanol, ethanol, propanol and butanol, at concentrations as low as 1 ppm at room temperature under UV excitation. It was found that resistive response decreases with the increasing carbon chain for aliphatic alcohols, as shown in Figure 2A. Also, the sensor response was diminished in the case of isomeric branching. Alcohols can remove adsorbed oxygen to release free electrons on SnO_2_, which promotes the rise of the photoconductivity in GaN NWs. This phenomenon contributed to achieve high sensitivity to alcohol vapors with a reasonable response-recovery time (200 s/100 s).

Aluri et al. [29] fabricated TiO_2_-Pt nanocluster-coated GaN NW sensor devices using a catalyst-free molecular beam epitaxy growth process. They reported that TiO_2_-Pt nanocluster-coated GaN NW hybrid sensor devices are able to detect as low as 100 ppb of ethanol and 0.5 ppm of methanol at room temperature under UV illumination with a short response-recovery time (80 s/60–80 s). They explained the alcohol selectivity of the sensor in terms of heat of adsorption, ionization energy and solvent polarity. In another work [28], a GaN NPs-based thick film sensor was able to detect 50 ppm of ethanol at room temperature with a fast response-recovery process (50 s/30 s). Recently, porous GaN nanorods were prepared by a hydrothermal method by Zhang et al. [48]. Gas-sensing measurements indicated that the porous sensor exhibits high sensitivity and strong selectivity to ethanol, and good stability at high temperature (360 °C). They presented a new route for the synthesis of GaN submicron rods [49]. The enhanced sensing performance toward ethanol is attributed to the large specific surface area, small grain size, and high length-to-diameter ratio of the developed GaN submicron rods.

The GaN nanostructured sensor synthesis method and corresponding sensing performance metrics including limit of detection, response/recovery times and operating temperatures are summarized in Table 2. It provides a brief comparative performance outline among different GaN nanostructure-based alcohol sensors reported in the past.

### 2.3. GaN Nanostructures-Based Methane Sensors

Having a lower explosion limit of 5.0%, methane detection is highly required in both the household and industrial arena [50]. Patsha et al. employed a CVD method in a vapor-liquid-solid process to make GaN NWs [51]. They introduced different amounts of oxygen in the NWs with varying oxygen impurity concentrations (10^5^ ppm, 10^3^ ppm, 10^2^ ppm and <2 ppm) in order to investigate the role of surface defects formed by the oxygen impurities in methane sensing. It was found that sensing response gradually decreases with the decrease of oxygen concentration in the GaN NWs. The prepared sensors were able to detect as low as 50 ppm of methane at 125 °C with moderate response-recovery time (90 s/100 s), as shown in Figure 2B. Also, it was observed that sensor response increases with the increase of temperature, which confirms the chemisorption instead of physisorption of CH_4_ molecules on the GaN nanowire surface [52]. The activation energy required for the chemisorption to occur during gas sensing is provided by the temperature rise.

Popa et al. [53] performed photo-electro-chemical (PEC) etching of GaN layers in a KOH solution and obtained pyramidal GaN layer structures. This sensor was able to detect 1% of methane gas at an operating temperature of 200 °C with a rapid response-recovery process (10 s/60 s). Moreover, it was found that PEC etching in a H_3_PO_4_-based solution facilitates nanoneedle structure formation and exhibits good sensing performance towards alcohols. Previously, GaN NWs incorporated with Au NPs had been reported as a promising methane sensor showing large conductivity changes on exposure to the target gas [54].

### 2.4. GaN Nanostructures-Based Benzene and Its Derivatives Sensors

GaN NWs have been functionalized with TiO_2_ nanoclusters, using RF magnetron sputtering, and employed UV excitation, to detect vapors of aromatic compounds [55]. This hybrid sensor device was able to detect as low as 50 ppb of benzene, toluene, ethylbenzene, xylene and chlorobenzene mixed with air at room temperature. The response and recovery times were quite short (60 s/75 s) because of the high reactivity of TiO_2_ nanoclusters. Figure 3A shows the variation of sensor current of the GaN/TiO_2_ device for 1000 ppm concentration of different aromatic compounds’ in the presence of UV light. For reference, the response of the device for air is also shown in the figure.

In another work, the same GaN/TiO_2_ hybrid sensor was implemented for the photo-enhanced detection of trinitrotoluene (TNT) and dinitrobenzene (DNB) at room temperature [56]. It detected as little as 500 ppt of TNT in air and DNB down to 10 ppb with a rapid sensor response-recovery process. Figure 3B illustrates the sensitivity profile of the synthesized device in presence of 1 ppm of different aromatics and nitro-aromatics. It is seen that the sensor is strongly selective to TNT against other interfering aromatic compounds. Being strongly electronegative, nitro-aromatics facilitate charge transfer between the adsorbed species on TiO_2_ NCs and the nitro groups, which in turn is reflected in sensing response. It was found that electron affinity increases with the increase in the number of nitro-groups bonded to the aromatics and thus the sensitivity is also enhanced. Table 3 provides a brief comparative performance outline among different GaN nanostructures-based methane, benzene and its derivatives sensors reported in the past.

### 2.5. GaN Nanostructures-Based O_2_, O_3_, NO and NO_2_ Sensors

Although GaN nanostructures have been very suitable for hydrogen and alcohol sensing, detection of various oxidizing gases was also demonstrated. P-i-n GaN nanorods (NRs), comprising of InGaN/GaN multi-quantum wells, have been reported recently for NO gas sensing [57]. This nanorods-based sensor was able to sense as low as 10 ppm of NO at room temperature. Though the response time was moderate (180 s), the recovery process of the device was too sluggish (400 s) even under UV illumination. Moreover, it was highly selective toward NO gas against other interfering oxidizing gases due to numerous surface states possessed by InGaN NRs. 

In another study, GaN nanowires were attached on pencil graphite electrodes using a hydrothermal method for NO detection [58]. The developed sensor allowed a wide detection range of 1.0 μM to 1.0 mM with a correlation coefficient of 0.999 and a detection limit of 0.180 µM, as obtained from cyclic voltammetry and amperometric measurements.

The ppb level detection of NO_2_ was demonstrated by titania (TiO_2_) nanoclusters-functionalized GaN submicron wire fabricated by a top-down approach [59]. The GaN/TiO_2_ sensor showed a lowest detection limit of 10 ppb of NO_2_ in air at room temperature (20 °C) (Figure 4A). The response/recovery (140/160 s) process was quite fast. It exhibited strong selectivity when exposed to NO_2_ and other interfering gases. Also, the device performance was degraded very little when exposed to siloxane for a one-month period. Previously, Shi et al. [60] fabricated hybrid gas sensors based on TiO_2_-decorated GaN nanowires for NO_2_ detection. The thickness and doping concentration of TiO_2_ were engineered to improve the transducer function. Results showed that stable n-type response was acquired for a doping range from 10^17^ cm^−3^ to 10^19^ cm^−3^. Maier et al. [61] designed an opto-chemical GaN/InGaN NW heterostructure and investigated its photoluminescence response for some oxidizing gases including O_3_, NO_2_, and O_2_ on it. The sensor system detected O_3_^−^, NO_2_^−^, and O_2_^−^ with a limit of detection (LoD) of 50 ppb, 500 ppb and 100 ppm, respectively at room temperature (Figure 4B–E). Previously, Pt-Pd alloy thin film had been incorporated onto suspended GaN NWs grown by VLS method [62]. The developed sensor exhibited good sensing response toward NO_2_ at a high operating temperature of 350 °C. The lowest detection limit was estimated as 100 ppm, with a reasonable response time. However, the sensor recovery process was slow (>100 s). 

### 2.6. GaN Nanostructures-Based SO_2_, H_2_S, NH_3_ and CO_2_ Sensors

In another work, GaN nanowires were developed on Si substrates using stepper lithography assisted dry-etching in a top-down fabrication approach [63]. The nanowires were functionalized by the deposition of different metal oxides—ZnO, WO_3_ and SnO_2—_using optimized RF sputtering. The ZnO/GaN sensor was found to be the best candidate for precise SO_2_ detection as shown in Figure 5A. It showed a response magnitude of 12.1% at a concentration of 10 ppm of SO_2_ in air at room temperature (20 °C) with a reasonable response/recovery (230/275 s) period. Additional sensor performance metrics such as adsorption and desorption rate, cross-sensitivity to interfering gases, and long-term stability at various environmental conditions were studied on the ZnO/GaN sensor device. In addition, the well-known cross-sensitive behavior of ZnO was resolved using principal component analysis (PCA). Chitara et al. [38] prepared a thick-film sensor based on GaN nanoparticles using a simple chemical route and investigated the sensing properties of NH_3_ and H_2_S gases at room-temperature. The variation of sensitivities for NH_3_ and H_2_S with the concentration of the vapor is shown in Figure 5B. It was observed that NH_3_ sensitivity is higher than 50% for 500 ppm. The response and recovery times for NH_3_ were found to be 200 s and 90 s, respectively, whereas it was 360 s and 150 s for H_2_S. 

Detection of CO_2_ gas utilizing metal-oxide-based sensors is challenging due to the chemical inertness and high stability of CO_2_ at room-temperature. Thomson et al. [64] fabricated GaN submicron wire-based chip-scale, low-power and nanoengineered chemiresistive gas-sensing architecture for CO_2_ detection. Their design utilized the selective adsorption properties of the nano-photocatalytic clusters of metal-oxides and metals. They achieved better selectivity for CO_2_ detection in high relative humidity conditions (Figure 5C). 

The GaN nanostructured sensor synthesis methods and the corresponding sensing performance metrics, including limit of detection, response/recovery times, and operating temperatures are summarized in Table 4. They provide a brief comparative performance outline among different GaN nanostructures-based oxidizing and reducing gas sensors reported to date.

## 3. Evaluation of Overall Sensor Performance

In order to be useful in real world applications, gas/chemical sensors need to possess strong response magnitude, fast response-recovery, excellent selectivity, long operating life and stable device performance. Unfortunately, a gas sensor with all these sensing properties has not been developed yet. In general, while some sensors show very high response magnitudes, their response-recovery processes are however quite slow. Some of them are promising in many aspects but not adaptive to large scale production. There is need for a common scale that we can use to compare the reported sensors to screen out optimal sensor devices. In this work, a novel metric, the product of response time and limit of detection, has been calculated for each sensor in order to quantify and compare the overall sensing performance of reported GaN nanostructures-based devices. The lower the value of the calculated metric, the faster the sensor is with a reasonable detection limit, which means it possesses better overall sensor performance. The obtained values of the proposed metric for different types of GaN nanostructures-based sensors are illustrated in Figure 6.

It is found that the InGaN/GaN NWs-based sensor exhibits the lowest multiplication value for H_2_ gas sensing (Figure 6A), whereas GaN/(TiO_2_–Pt) NWNC-based sensor shows the lowest value for ethanol sensing (Figure 6B). Also, it turns out that GaN/TiO_2_ NWNC is well suited for TNT sensing, as shown in Figure 6C. The more sensors taken in the comparison, the more accurate the suggestion of the optimal sensor will be.

## 4. Photo-Assisted Gas Sensing with GaN Nanostructures

Generally, heated MEMS sensors require power in the mW range for operation [65]. For irradiating nanostructure-based sensors, the typical power consumed by a UV diode is in µW range, which is quite low as compared to that of heated sensors [66]. Photo-enabled sensing makes it possible to operate gas sensors at room-temperature, resulting in a significant reduction in operating power demand. Typically, metal oxide-based gas sensors are operated at elevated temperatures in order to enhance their surface reactivity through a redox reaction [67]. GaN nanostructures with surface functionalization have the ability to incorporate photo-assisted gas sensing [68]. When UV light is illuminated with a greater energy than GaN and metal oxide bandgaps, electron–hole pairs are generated in them. Holes diffuse toward the GaN surface because of surface band bending. Thus, carrier lifetime and photocurrent increase within the GaN. The dynamic active adsorption sites are generated on the sensor surface through photo-desorption, which allows the sensors to achieve room temperature gas sensing. In another study, it was observed that the photogenerated electrons facilitate oxygen adsorption and produce the photoinduced oxygen ions [69]. These ions contribute to the room-temperature gas detection and enhanced sensor performance. In addition, UV light improves sensor response and recovery rate along with sensitivity. The wavelength and intensity of UV light had been found to have an impact on the adsorption/desorption kinetics of gas molecules [70]. Photo-assisted sensing also helps to withdraw the residual gas molecules that build up with each exposure. Thus, the absorption sites are fully recovered, and the sensor is able to return to its original baseline resistance.

## 5. Machine Learning Algorithms on GaN Nanostructured Sensors 

Machine learning has proven to be an excellent method of data classification and processing. Learning algorithms are suitable for classifying and calibrating gas sensors based on the measurement data received from the sensors. Recently, a gas sensor array was reported comprising of GaN nanowires functionalized with metal incorporated TiO_2_ and ZnO, as displayed in Figure 7A [71]. The sensor array was tested with NO_2_, ethanol, SO_2_ and H_2_ in presence of H_2_O and O_2_ gases in both unmixed and mixed conditions at room temperature. Gas analytes leave footprints on the array, which are analyzed to identify the cross-sensitive gases. Various supervised machine learning algorithms including decision tree, support vector machine, naive Bayes and k-nearest neighbor were trained and tested for the classification of gas type. It is seen from the results that the support vector machine and naive Bayes classifiers show better classification accuracy than all other models. Furthermore, unsupervised principal component analysis (PCA) was utilized on the response patterns obtained from the array. Results indicate that all the individual gases form discrete clusters in the score plot, exhibiting an enhanced analyte selectivity (Figure 7B).

Non-linearity is observed in the gas responses when gases remain in mixed conditions [72]. Artificial neural networks (ANNs) have been highly efficient to capture the non-linear response pattern of gas mixtures. In another study, various artificial neural network (ANN) algorithms were trained and tested for the identification and quantification of gas mixtures based on GaN nanowires [73]. A back-propagation neural network model was found to be the optimal classifier among all the considered ANN algorithms based on the statistical and computational complexity results. Furthermore, concentrations of the labelled gases had been predicted in ppm based on the optimal model. 

## 6. Molecular Simulation of GaN-Based Gas Sensors

Several efforts have been made to investigate the adsorption properties of GaN toward different chemical and gas molecules by first-principle method calculations using density functional theory (DFT) as shown in Table 5. Yong et al. [74] studied the adsorption of gas molecules such as- SO_2_, NO_2_, HCN, NH_3_, H_2_S, H_2_, CO_2_, H_2_O on the graphitic GaN sheet (PL-GaN). Results indicated that SO_2_ and NH_3_ gas molecules were chemisorbed on the PL-GaN sheet and strong modifications in electronic structures were observed after their adsorption (Figure 8). The local DOS of gas molecules are presented by the dark yellow filled area under the DOS curve. The positive and negative DOS values indicate spin-up and spin-down states, respectively. Furthermore, the adsorption of NO and NO_2_ molecules introduced spin polarization in the PL-GaN sheet, indicating that it can be employed as a magnetic gas sensor for NO and NO_2_ sensing. In another study, adsorption properties such as- adsorption energy, adsorption distance, Hirshfeld charge, electronic properties, and recovery time were investigated for NO, NH_3_, and NO_2_ gas molecules on two-dimensional GaN with a tetragonal structure (T-GaN) [75]. It was found that electronic structures within TGaN exhibited significant changes when NO_2_ and NO were adsorbed. On the other hand, electronic structures remained almost unchanged due to NH_3_ adsorption on TGaN.

It is well known that wurtzoids are bundles of capped (3, 0) nanotubes that form the wurtzite phase when they reach nanocrystal or bulk sizes. Abdulsattar et al. [76] performed DFT computations and reported that GaN wurtzoids as a representative of GaN nanocrystals are suitable for hydrogen sensing nanostructures. The N sites were found to be responsible for the H sensing capability. Ga sites were either strongly bonded to other air gases or have higher interaction energy (Van der Waals’ forces) with H molecules that made them highly stable. Furthermore, structural and vibrational properties for bare and hydrogen passivated GaN molecules had been computed and compared with the experimental bulk values.

Recently, molecular models of metal oxide-coated GaN nanostructures have been simulated to study their adsorption and electronic properties for gas sensing applications [77,78]. Results indicated that TiO_2_ functionalization enabled the most energy favorable surface for NO_2_ adsorption among the considered metal oxides. The total density of states (TDOS) and projected density of states (PDOS) of the nanostructures have been calculated and compared, revealing the nature and strength of chemical interaction between the orbitals of gas molecule and sensor surface. It was also found that recovery of the gas sensing process gets slower because of larger chemical stability of an adsorption system. Therefore, both strong and weak chemical bindings between gas and sensor surface hamper response/recovery speed. The impact of humidity on the adsorbate-sensor interaction has been revealed as well.

## 7. Gas Sensing Mechanisms

It is very important to understand the gas/chemical sensing mechanism of GaN nanostructure-based sensor devices for optimizing the performance of the devices using novel functionalization schemes. The fundamental mechanism responsible for a change in sensor conductivity is the trapping of electrons at adsorbed molecules and band bending induced by these charged molecules [79]. Here, a brief discussion on the sensing mechanism of GaN nanostructured sensors is given for the case of metal-oxide functionalized GaN nanowires at room temperature. Aluri et al. reported that electron-hole pairs are generated in both GaN backbone and metal-oxide upon UV illumination [55]. These photogenerated holes in the nanowire tend to diffuse toward the surface due to the surface band bending. The chemisorbed oxygen molecule (O^−2^) and hydroxide ions (OH^−^) capture a hole and desorb, creating a surface defect active site as shown in Figure 9A,B. Analyte molecules chemisorb at those active sites and cause surface potential modification of the GaN backbone through dynamic trapping and de-trapping of charge carriers. As a result, the sensor current got modulated in proportion to the analyte concentration. In another work, Zhong et al. [42] investigated the H_2_ sensing mechanism on Pt/GaN, where a Schottky barrier is generated between Pt and GaN due to the energy difference between their work function and electron affinity (Figure 9C). The analyte H_2_ molecules are dissociated by the Schottky contact Pt. Then, the chemisorbed H atoms are diffused to the Pt-GaN interface and absorbed by the absorption sites. Since there is a built-in electric field in the depletion region, dipoles are formed [80]. In consequence, the Schottky barrier height is lowered, contributing to a decrease in Schottky diode voltage. It has been revealed that the magnitude and direction of charge flow rely on the device work function, highest occupied molecular orbital (HOMO), and lowest unoccupied molecular orbital (LUMO). Moreover, chemisorption-induced energy, chemical potentials, electron affinity, and ionization potential have an impact on electron redistribution between the gas molecules and adsorption systems. If the energy difference between Fermi energy and LUMO is much less compared to HOMO, electrons are most likely to transfer to LUMO of the analyte gas. Here, the electrons are transferred from the sensor to the gas molecule by the process of quantum tunneling [81]. As a consequence, the Fermi energy of the device starts going down toward valence band. The charge transfer continues to take place until equilibrium Fermi energy is reached within the adsorption system.

## 8. GaN Sensors in Internet of Things (IoT) Applications

It is well known that IoT is a large range of devices that connect to a network and have ability to transfer data to another connected device. IoT devices are required to operate under harsh environmental conditions and exhibit long lifetime [82]. Being chemically robust in nature, semiconducting GaN has the advantages of radiation-tolerance and robustness to environmental temperature variation [83]. It exhibits stable operation across a wide temperature and humidity range. Though GaN shows very little cross-sensitivity toward water vapor [84], its functionalizing material such as metal oxide is cross-sensitive to ambient humidity [85,86,87]. Therefore, the variation of humidity in real world application causes degradation in sensitivity and selectivity of cross-sensitive sensors. Techniques like sensor array had been employed to enhance the humidity-affected sensor performance through pattern-based sensing [88]. Overall, GaN-based sensors are less cross-sensitive to humidity than the most explored metal oxide-based gas sensors.

GaN-based gas sensors are capable of providing the characteristics required by IoT platforms, thus they are suitable candidate for the IoT applications such as remote air quality monitoring. They can be also incorporated into a multi-purpose field surveillance robot which uses multiple IoT cloud servers [89]. GaN sensors have the capability to be utilized in wireless sensor networks for toxic gas boundary area detection in large-scale petrochemical plants [90]. Furthermore, they can offer high performance sensing in IoT-based vehicle emission monitoring systems [91]. Since GaN sensor devices have the advantages of ultra-low power operation, they can be integrated into embedded-chip or plug-in module as well.

## 9. Conclusions and Future Perspectives

This work reviews and categorizes the progress in GaN nanostructures-based sensors for detection of gas/chemical species such as hydrogen (H_2_), alcohols (R-OH), methane (CH_4_), benzene and its derivatives, nitric oxide (NO), nitrogen dioxide (NO_2_), sulfur-dioxide (SO_2_), ammonia (NH_3_), hydrogen sulfide (H_2_S) and carbon dioxide (CO_2_). The standard sensing performance parameters like limit of detection, response/recovery time and operating temperature for different types of sensors and structures were summarized comprehensively for the comparative study. The proposed metric, product of response time and limit of detection, has been calculated for each sensor to measure and compare the overall sensing performance among reported GaN nanostructures-based devices so far. Based on the analysis of sensing characteristics and the proposed metric, it was found that InGaN/GaN NW sensor shows superior overall sensing performance for H_2_ gas sensing. Also, GaN/(TiO_2_–Pt) and GaN/TiO_2_ NWNC sensors are highly suitable for ethanol and TNT sensing, respectively. Moreover, metal-oxide coated GaN NWs exhibit reliable sensing performance toward various oxidizing gases including NO_2_ and SO_2_. Theoretical studies on molecular models of gas molecules and GaN have been reviewed. Furthermore, a brief analysis of the implementation of machine learning on GaN nanostructured sensors and sensor array has been presented. In addition, gas sensing mechanisms of the GaN sensors have been discussed. This overview on the GaN nanostructures-based gas sensors is helpful for the researchers to gain a quick understanding of the status of GaN nanostructure-based sensors.

There are still many challenges in terms of sensitivity, selectivity, response/recovery speed, and reliability, which need to be addressed to obtain the desired gas sensors. Semiconducting GaN is one of the most promising materials to address these shortcomings in sensing applications. Though most of the reported GaN nanostructure-based gas sensors show promising performance, research efforts are needed to address important issues like device scalability, reproducibility, and reliability as well. Also, researchers performed their gas sensing experiments in an ideal laboratory condition, which does not simulate real-world environment. Therefore, comprehensive real-world field testing needs to be done for the sensor devices to be effective and long lasting in real world application.

## Figures and Tables

**Figure 1 sensors-20-03889-f001:**
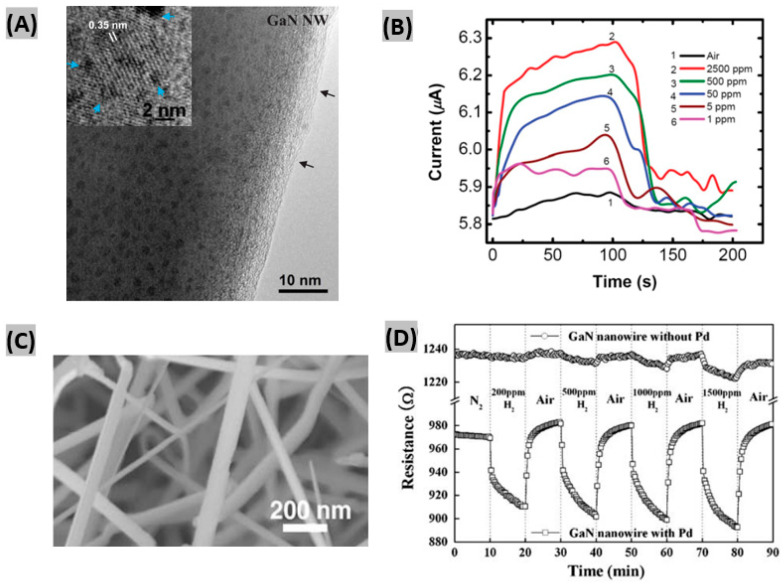
(**A**) HRTEM image of TiO_2_ sputtered GaN NW after Pt deposition. Inset shows a magnified image where blue arrows indicate Pt clusters deposited on TiO_2_. (**B**) Variation of sensor current of GaN/(TiO_2_-Pt) device toward various concentrations of H_2_ in nitrogen. Figures adapted with permission from [29], Copyright 2012 IOP Publishing Ltd. (**C**) Scanning electron microscopy (SEM) micrographs of as-grown nanowires. (**D**) Resistance responses of Pd-coated and uncoated GaN NW at the exposure of H_2_ concentrations ranging from 200 ppm to 1500 ppm. Figures adapted with permission from [36], Copyright 2008 TMS.

**Figure 2 sensors-20-03889-f002:**
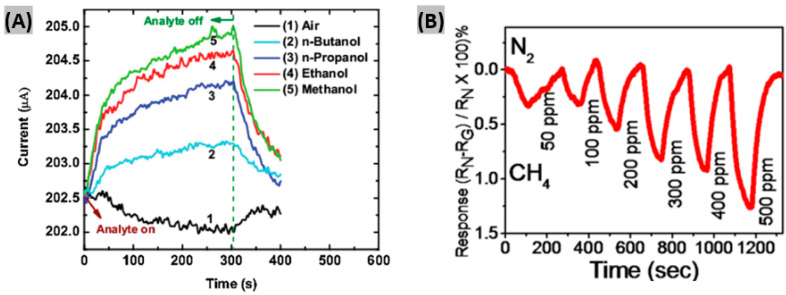
(**A**) Variation of sensor currents of the SnO_2_ functionalized GaN NW device toward 500 ppm of various alcohols. Figure adapted with permission from [47], Copyright 2012 Elsevier. (**B**) Sensor responses of GaN NW at the exposure of CH_4_ gas concentrations ranging from 50 ppm to 500 ppm. Figure adapted with permission from [51], Copyright 2015 American Chemical Society.

**Figure 3 sensors-20-03889-f003:**
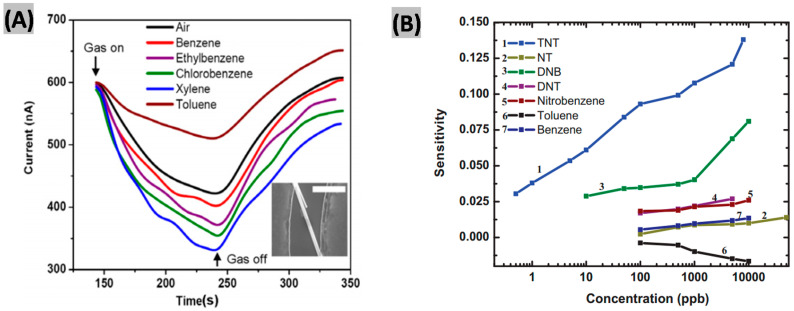
(**A**) Variation of sensor current of the GaN/TiO_2_ device toward 1000 ppm of different aromatic compounds’ vapor with a reference of air response in the presence of UV light. Figures adapted with permission from [55], Copyright 2011 IOP Publishing Ltd. (**B**) Sensitivity profile of the synthesized device in presence of 1 ppm of different aromatics and nitro-aromatics. Figures adapted with permission from [56], Copyright 2013 IEEE.

**Figure 4 sensors-20-03889-f004:**
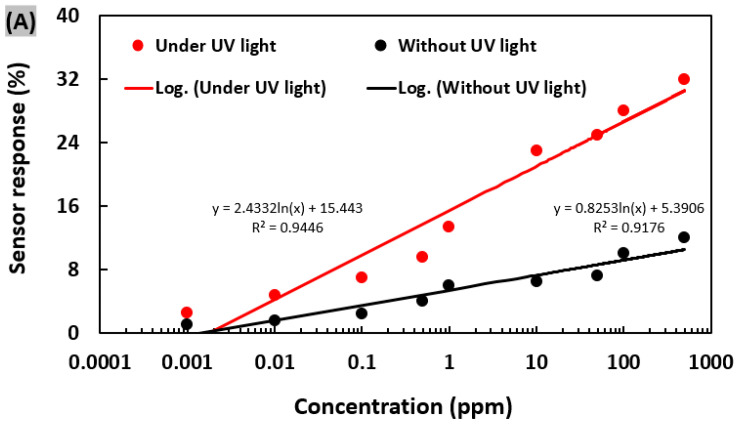
(**A**) Response fitting curves of the of GaN/TiO_2_ NW sensor to NO_2_ concentrations ranging from 1 ppb to 500 ppm with UV light and without UV light at room temperature (20 ºC). Figure adapted with permission from [59], Copyright 2020 IOP Publishing Ltd. (**B**) Schematic representation of the GaN/InGaN NW on Si substrate. Variation of (**C**) O_2_-response, (**D**) NO_2_-response, and (**E**) O_3_-response for GaN/InGaN NW sensor with gas concentration and operating temperature. Figures adapted with permission from [61], Copyright 2014 Elsevier.

**Figure 5 sensors-20-03889-f005:**
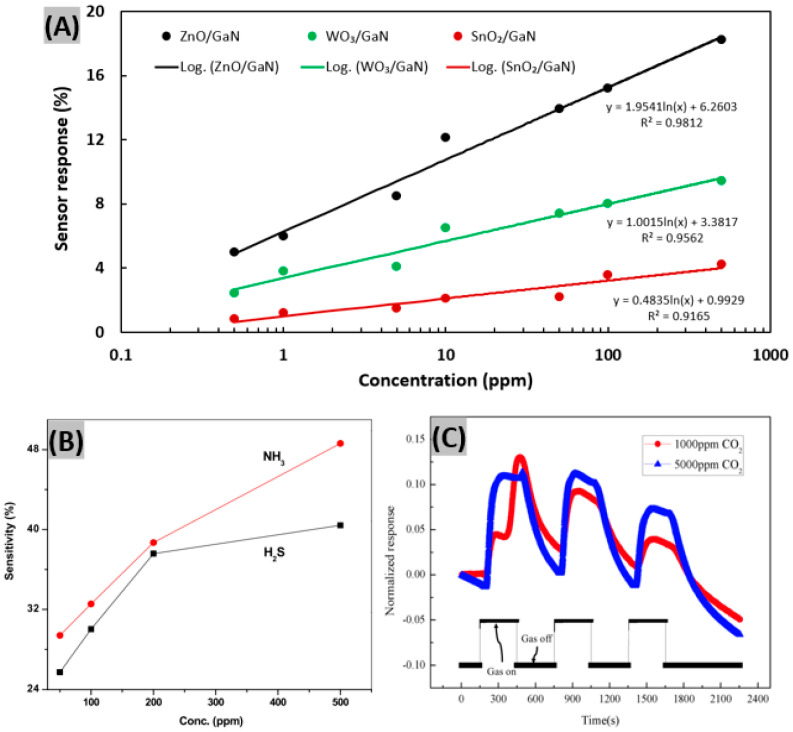
(**A**) SO_2_ gas response fitting lines for the ZnO/GaN NW sensor (black), WO_3_/GaN NW sensor (green) and SnO_2_/GaN NW sensor (red) under UV light at room temperature (20 °C). Figure adapted with permission from [63], Copyright 2020 Elsevier. (**B**) The plotting of sensitivity variation of GaN nanoparticles for H_2_S and NH_3_ gas at 300 K. Figure adapted with permission from [38], Copyright 2010 Elsevier Ltd. (**C**) Normalized responses of metal-oxide coated GaN NW sensor device for 1000 and 5000 ppm of CO_2_ gas under UV light. Figure adapted with permission from [64], Copyright 2018 ICES.

**Figure 6 sensors-20-03889-f006:**
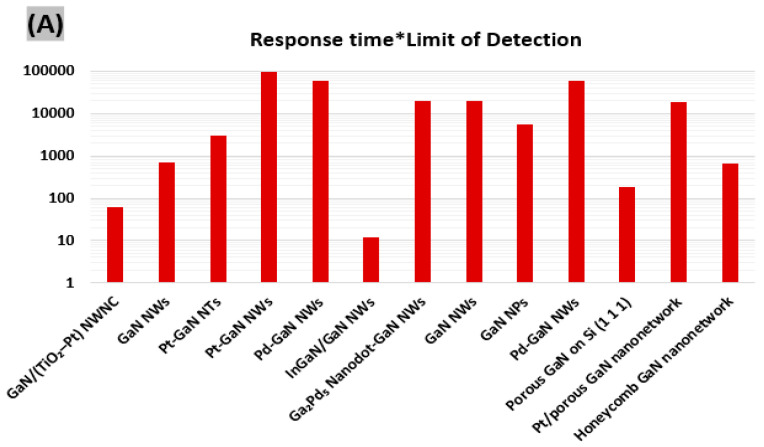
Comparison of the product of response time and limit of detection value among previously reported GaN nanostructures-based sensors for (**A**) H_2_, (**B**) alcohols, and (**C**) other gases.

**Figure 7 sensors-20-03889-f007:**
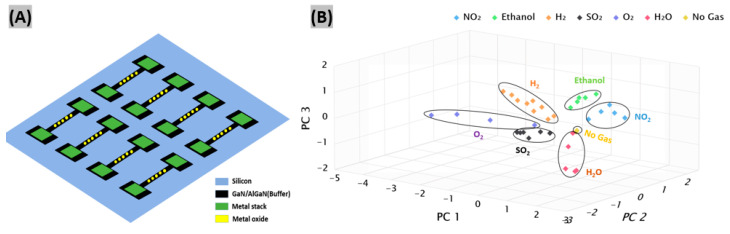
(**A**) The schematic representation of a sensor array comprising of eight metal/metal-oxide coated GaN NWs. (**B**) Score plot from principal component analysis (PCA) analysis for various concentrations of NO_2_, ethanol, SO_2_, H_2_, O_2_ and H_2_O, which includes up to 95.1% of the total variance. Figures adapted with permission from [71], Copyright 2020 IEEE.

**Figure 8 sensors-20-03889-f008:**
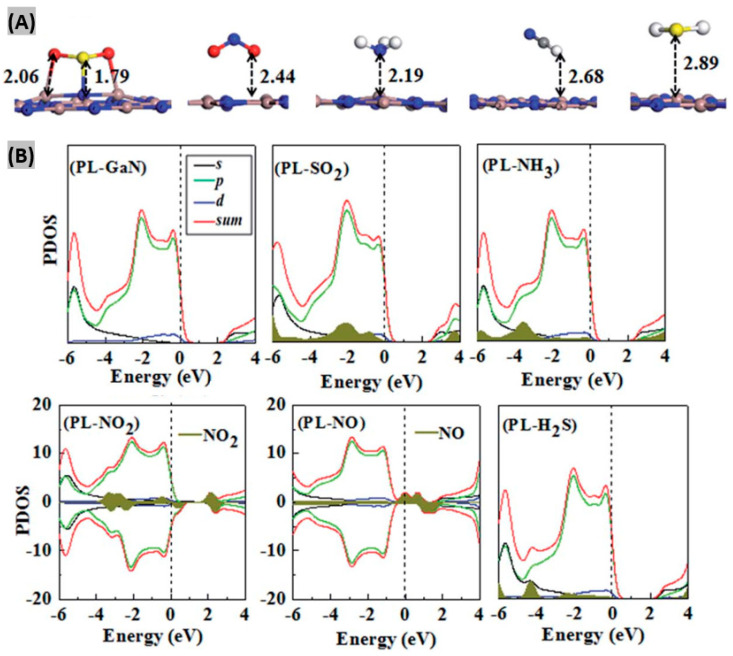
(**A**) The most stable structures of the PL-GaN sheet with gas molecule adsorption, (From left) SO_2_, NO_2_, NH_3_, HCN and H_2_S. Adsorption distances are presented in Å. Ga, N, S, O, C and H atoms are brown, blue, yellow, red, grey and white, respectively. (**B**) Total and partial density of states (DOS) of the most stable structures of the PL-GaN sheet and its adsorption systems with SO_2_, NH_3_, NO_2_, NO, and H_2_S. Fermi-level energy is indicated by vertical dashed line. Figures adapted with permission from [74], Copyright 2017 The Royal Society of Chemistry.

**Figure 9 sensors-20-03889-f009:**
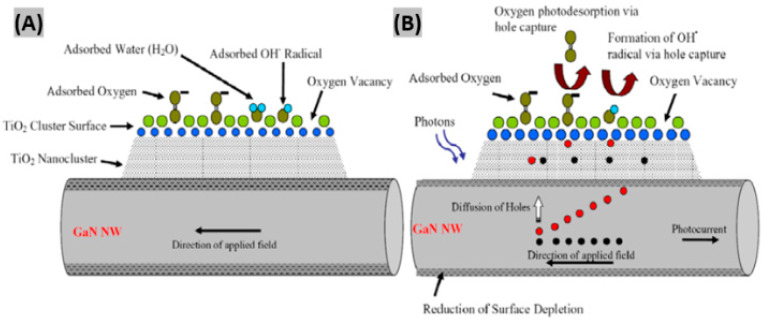
Gas sensing mechanism illustration using schematic representation of the GaN/TiO_2_ sensor (**A**) in the dark, and (**B**) under UV illumination. Figures adapted with permission from [55], Copyright 2011 IOP Publishing Ltd. (**C**) Sensing mechanism of H_2_ gas on GaN/Pt sensor device. Figure adapted with permission from [42], Copyright 2014 Hydrogen Energy Publications, LLC. Published by Elsevier Ltd.

**Table 1 sensors-20-03889-t001:** Synthesis and sensing properties of GaN nanostructures based H_2_ sensors.

Sensor Type	Method of Synthesis	Response/Recovery Times	Lowest Detection Limit	Operating Temperature	Reference
Ga_2_Pd_5_ Nanodot-GaN NWs	Chemical vapor deposition	200/800 s	100 ppm	RT	[27]
GaN/(TiO_2_–Pt) NWNC	Catalyst-free molecular beam epitaxy	60/45 s	1 ppm	RT (UV)	[29]
GaN NWs	Chemical vapor deposition	103/82 s	7 ppm	RT	[31]
InGaN/GaN NWs	Plasma-assisted molecular beam epitaxy	1/1 min 10/10 min	200 ppb 10 ppm	80 °C RT	[32]
Pt-GaN NWs	Chemical vapor deposition	500 s/120 s	200 ppm	RT	[33]
Pt-GaN NTs	Chemical vapor deposition	120 s/5 min	25 ppm	100 °C	[35]
Pd-GaN NWs	Catalytic chemical vapor deposition (CVD)	5/2 min	200 ppm	RT	[36]
Pt/porous GaN nanonetwork	Molecular beam epitaxy	1/8 min	320 ppm	RT	[37]
GaN NPs	Solvothermal decomposition	110/70 s	50 ppm	RT	[38]
Pd-GaN NWs	Thermal chemical vapor deposition	300/120 s	200 ppm	RT	[39]
GaN NWs	Two step CVD	100/120 s	200 ppm	RT	[40]
Porous GaN on Si (1 1 1)	Plasma-assisted molecular beam epitaxy	3/2.5 min	1 sccm	RT	[41]
Honeycomb GaN nanonetwork	Molecular beam epitaxy	13.2/- s 4.4/- s	50 ppm	30 °C 100 °C	[42]

**Table 2 sensors-20-03889-t002:** Synthesis and sensing properties of GaN nanostructures-based alcohol sensors.

Sensor Type	Target Analyte	Synthesis Method	Response/Recovery Times	Lowest Detection Limit	Operating Temperature	Reference
GaN/(TiO_2_–Pt) NWNC	Ethanol	Catalyst-free molecular beam epitaxy	80/60 s	100 ppb	RT (UV)	[29]
GaN/(TiO_2_–Pt) NWNC	Methanol	Catalyst-free molecular beam epitaxy	80/80 s	0.5 ppm	RT (UV)	[29]
GaN NPs	Ethanol	Solvothermal decomposition	50/30 s	50 ppm	RT	[38]
GaN/Si- nanoporous pillar array (NPA)	Methanol	Chemical vapor deposition	8/7 s	5 ppm	350 °C	[44]
Porous GaN-NFs	Ethanol	Electrospinning	8/5 s	50 ppm	320 °C	[45]
SnO_2_/GaN NW	Alcohol vapors	Catalyst-free molecular beam epitaxy	200/100 s	1 ppm	RT (UV)	[47]
GaN submicron rods	Ethanol	Hydrothermal method	2/42 s	5 ppm	360 °C	[49]

**Table 3 sensors-20-03889-t003:** Synthesis method and sensing properties of GaN nanostructures-based methane, benzene and its derivatives sensors.

Sensor Type	Target Analyte	Fabrication Technique	Response/Recovery Times	Lowest Detection Limit	Operating Temperature	Reference
GaN NWs	Methane	CVD in vapor-liquid-solid process	90/100 s	50 ppm	125 °C	[51]
GaN-based two-sensor array	Methane	Metalorganic CVD with photoelectron-chemical etching	10/60 s	1%	200 °C	[53]
GaN–TiO_2_ hybrid NWNC	Benzene, toluene, ethylbenzene, xylene and chlorobenzene	Catalyst-free molecular beam epitaxy	60/75 s	50 ppb	RT (UV)	[55]
GaN/TiO_2_ NWNC	TNT Dinitrobenzene (DNB)	Catalyst-free molecular beam epitaxy	30/30 s	500 ppt 10 ppb	RT (UV)	[56]

**Table 4 sensors-20-03889-t004:** Synthesis method and sensing properties of GaN nanostructures-based various oxidizing and reducing gas sensors.

Sensor Type	Target Analyte	Fabrication Technique	Response/Recovery Times	Lowest Detection Limit	Operating Temperature	Reference
GaN NPs	NH_3_	Simple chemical route	200/90 s	50 ppm	RT	[38]
GaN NPs	H_2_S	Simple chemical route	360/150 s	50 ppm	RT	[38]
p-i-n GaN NRs	NO	Plasma-assisted molecular beam epitaxy	180/400 s	10 ppm	RT (UV)	[57]
GaN/TiO_2_ NW	NO_2_	Stepper lithography assisted ICP etching	140/160 s	10 ppb	RT (UV)	[59]
GaN/InGaN NWs	O_2_ NO_2_ O_3_	Plasma-assisted molecular beam epitaxy	-/-	100 ppm 500 ppb 50 ppb	RT	[61]
Pt–Pd/GaN NWs	NO_2_	Vapor–liquid–solid (VLS) process	100/>100 s	100 ppm	350 °C	[62]
GaN/ZnO NW	SO_2_	Stepper lithography assisted ICP etching	230/275 s	10 ppm	RT (UV)	[63]
GaN/Metal-oxide NW	CO_2_	ICP etching	100/300 s	1000 ppm	RT (UV)	[64]

**Table 5 sensors-20-03889-t005:** The adsorption energy (eV), shortest adsorption distance (Å) and charge transfer (e) between gas molecules and GaN sensors at the most stable adsorption configuration based on DFT.

Materials	Target Gas	Adsorption Energy (eV)	Shortest Adsorption Distance (Å)	Charge Transfer (e)
Graphitic GaN sheet [74]	NO_2_	−0.493	2.44	−0.081
Graphitic GaN sheet [74]	SO_2_	−1.06	1.79	−0.209
Graphitic GaN sheet [74]	H_2_S	−0.446	2.89	0.139
2D Tetragonal GaN [75]	NO_2_	−0.673	2.066	−0.108
2D Tetragonal GaN [75]	NH_3_	−1.317	2.089	0.289
2D Tetragonal GaN [75]	NO	−0.872	1.374	−0.271
GaN wurtzoid [76]	H_2_	−0.025	~1	-
TiO_2_ coated GaN [77]	NO_2_	−2.31	0.25	0.214
ZnO coated GaN [77]	NO_2_	−1.96	0.28	0.187
SnO_2_ coated GaN [77]	NO_2_	−1.95	0.30	0.093

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
