# Peer review of "Gallium Nitride (GaN) Nanostructures and Their Gas Sensing Properties: A Review"

_sensors, 2020, doi:10.3390/s20143889_

Round 1

Reviewer 1 Report

This paper is a focused review in the gas sensing applications of GaN nanomaterials. GaN materials have attracted remarkable research interest in the last years and this focused review seems timely and of interest to the sensors community. There are, however, a few important aspects that Authors need to consider in a revised version of their paper. These are as follows:

  1. The different tables shown in the paper are aimed at comparing the performance of sensors employing GaN. In most cases it is indicated that sensors are operated at room temperature (RT). However, in many cases RT operation is accompanied by UV light irradiation of the material. The use or not of UV light irradiation of the sensor to promote response and/or recovery should be clearly stated in these tables.
  2. One of the most important performance parameters for a gas sensor is sensitivity (sensitivity would be defined as the slope of a calibration curve in which sensor response is plotted as a function of the concentration of a target gas or vapor). I strongly believe that the sensitivity parameter should be added to the different tables shown. That would perfectly complement the information given by the LOD and response time, etc.
  3. Related to my previous point. The figure of merit described in the paper could be improved as follows : FM = (response time *LOD)/sensitivity
  4. UV light is also power hungry, e.g. if a UV diode is used to irradiate the sensor. What is the power consumed (typical value) of a UV irradiated sensor? How this compares to heated MEMS sensors that typically need few 10 mW? A short discussion seems imperative.
  5. No information is given on the water vapor cross-sensitivity effect for GaN materials. Since the main applications would require the detection of target species in a real environment, reporting results and discussing moisture cross-sensitivity seems imperative.
  6. What is the long term stability of these materials in gas sensing? These aspects should be discussed as well.
  7. How GaN compares to well-known metal oxide nanomaterials? It would be good to compare the performance of GaN sensors to the one of metal oxides (that have an important share of the market in gas sensing) and to stress in which aspects GaN based devices show potential for replacing them. This would help the interested reader to better assess usability and interest.
  8. The paper is structured in 9 sections, some of which are rather short. Authors should consider merging some of the sections together.
  9. The English needs some minor improvements.

Author Response

Dear Editor and Reviewers,

We appreciate the thoughtful comments and suggestions regarding our manuscript entitled “Gallium Nitride (GaN) Nanostructures and Their Gas Sensing Properties: A Review” by Md Ashfaque Hossain Khan and Mulpuri V. Rao, submitted for publication in Sensors journal. We have significantly benefited from your comments and improved our manuscript accordingly. The revised parts in the article are highlighted. The revised manuscript is hereby submitted for your consideration.

Please contact us if you have any further questions.

Sincerely,

Md Ashfaque Hossain Khan

Electrical and Computer Engineering

George Mason University

4400 University Drive, Fairfax, Virginia 22030

Ph: 571-373-9991

Response to Reviewer 1

Thank you very much for reviewing our manuscript. We greatly appreciate you for the complimentary comments and suggestions. The followings are our point-by-point responses:

Comment: “The different tables shown in the paper are aimed at comparing the performance of sensors employing GaN. In most cases it is indicated that sensors are operated at room temperature (RT). However, in many cases RT operation is accompanied by UV light irradiation of the material. The use or not of UV light irradiation of the sensor to promote response and/or recovery should be clearly stated in these tables.”

Response: Thanks for pointing this out. In the revised manuscript, we have added the information of UV light irradiation in the cases of RT operations within all the Tables, (Table 1 - Page 4, Table 2 - Page 6, Table 3 - Page 7, and Table 4 - Page 12).

Comment: “One of the most important performance parameters for a gas sensor is sensitivity (sensitivity would be defined as the slope of a calibration curve in which sensor response is plotted as a function of the concentration of a target gas or vapor). I strongly believe that the sensitivity parameter should be added to the different tables shown. That would perfectly complement the information given by the LOD and response time, etc.”

Response: Initially we tried to collect and present the sensitivity values for the sensors given in the tables. Unfortunately, many sensors show nonlinear trend in the response vs concentration plot, i.e., they exhibit different sensitivity at different concentration regions. Also, their target concentration ranges were not alike. As a result, we could not present a single sensitivity value for each sensor in the tables.

Comment: “Related to my previous point. The figure of merit described in the paper could be improved as follows : FM = (response time *LOD)/sensitivity.”

Response: The limitation of finding a single sensitivity value for a sensor in many cases has been described in the previous point.

Comment: “UV light is also power hungry, e.g. if a UV diode is used to irradiate the sensor. What is the power consumed (typical value) of a UV irradiated sensor? How this compares to heated MEMS sensors that typically need few 10 mW? A short discussion seems imperative.”

Response: For irradiating nanostructure-based sensors, the typical power consumed by a UV diode is 100-500 uW, as found from the literature. It is quite less as compared to the heated MEMS sensors. There is a discussion on this topic in section 4, and we have further added the following sentences to that description, (Page 14, Lines 312-314) àGenerally, the heated MEMS sensors require power in mW range for operation [65]. For irradiating nanostructure-based sensors, the typical power consumed by a UV diode is in µW range, which is quite low as compared to the heated sensors [66].

Comment: “No information is given on the water vapor cross-sensitivity effect for GaN materials. Since the main applications would require the detection of target species in a real environment, reporting results and discussing moisture cross-sensitivity seems imperative.”

Response: Thanks for pointing this out. In the revised manuscript, we have added the following description focusing on the water vapor cross-sensitivity, (Page 18, Lines 436-441) àThough GaN shows very little cross-sensitivity toward water vapor [84], its functionalizing material such as metal oxide is cross-sensitive to ambient humidity [85]–[87]. Therefore, the variation of humidity in real world application causes degradation in sensitivity and selectivity of cross-sensitive sensors. Techniques like sensor array had been employed to enhance the humidity-affected sensor performance through pattern-based sensing [88]. Overall, GaN-based sensors are less cross-sensitive to humidity than the most explored metal oxide-based gas sensors.

Comment: “What is the long term stability of these materials in gas sensing? These aspects should be discussed as well.”

Response: It has been observed that almost all the reported GaN sensors show stable performances over long period. Therefore, in the description and tables, we have mainly focused on the primary sensor metrics such as response magnitude, response/recovery times, limit of detection, operating conditions, and fabrication methods.

Comment: “How GaN compares to well-known metal oxide nanomaterials? It would be good to compare the performance of GaN sensors to the one of metal oxides (that have an important share of the market in gas sensing) and to stress in which aspects GaN based devices show potential for replacing them. This would help the interested reader to better assess usability and interest.”

Response: Thanks for pointing this out. In the revised manuscript, we have added the following sentences describing the comparison between metal-oxide and GaN sensors, (Page 2, Lines 55-61) àAlthough commercially available metal-oxide nanostructure-based gas sensors show high sensitivity and low detection limit [22], they suffer from issues such as poor analyte selectivity, high operating temperature and unstable performance in harsh environments [23]. GaN nanostructures offer stable operation under various radiations and in space condition. They operate at room-temperature can also tolerate large variations of temperature and humidity as compared to metal-oxides [24]. Thus, GaN nanostructure sensors have the potential to take over a significant share of the market in gas sensing.

Comment: “The paper is structured in 9 sections, some of which are rather short. Authors should consider merging some of the sections together.”

Response: Now after the revision, smaller sections look unique and unrelated to each other. Therefore, they are kept as separate sections.

Comment: “The English needs some minor improvements.”

Response: The whole manuscript has been thoroughly revised and checked grammatically. English of the manuscript has been further improved.

Reviewer 2 Report

The authors of the article with title “Gallium Nitride (GaN) Nanostructures and Their Gas Sensing Properties: A Review” study the GaN nanostructures prepared by different deposition methods with introducing of different doping levels. In the review were presented different GaN based nanostructures such as nanowires, nanorods, nanotubes, nanoparticles and nanobelts. The sensing performance of it, like limit of detection, response/recovery time and operating temperature were used to compare different type of sensors with different structures. In the paper were presented wide range of gases which were detected such as hydrogen (H2 ), alcohols (R-OH), methane (CH4 ), benzene and its derivatives, nitrogen monoxide (NO), nitrogen dioxide (NO2 ), sulfur-dioxide (SO2 ), ammonia (NH3 ), hydrogen sulfide (H2S) and carbon dioxide (CO2 ).

After the structured description part, followed part is titled Evaluation of Overall Sensor Performance”. There, the authors simply described usual gas sensor behavior and gives to the readers some useful advice. In current review, photo-assisted sensing is included as well. Shortly it has been described the mechanism of sensing enhancement as an effect from the UV-light.

In this review paper, some theoretical aspects were described: machine learning algorithms, some molecular simulations and gas sensing mechanisms.

The study is written using simple and understandable style. Тhe importance of the idea to popularize necessity of structured GaN sensors and their advantages is still valid. However, for improving the quality of the study, a question emerge.

Question:

1. The conclusions are very general. Every review part must have the most important conclusion including concrete example or more general trend or direction in gas sensing. Authors must show the most suitable GaN based sensors in one or many cases. It is not easy to follow/understand the author’s main message in conclusions.

Author Response

Dear Editor and Reviewers,

We appreciate the thoughtful comments and suggestions regarding our manuscript entitled “Gallium Nitride (GaN) Nanostructures and Their Gas Sensing Properties: A Review” by Md Ashfaque Hossain Khan and Mulpuri V. Rao, submitted for publication in Sensors journal. We have significantly benefited from your comments and improved our manuscript accordingly. The revised parts in the article are highlighted. The revised manuscript is hereby submitted for your consideration.

Please contact us if you have any further questions.

Sincerely,

Md Ashfaque Hossain Khan

Electrical and Computer Engineering

George Mason University

4400 University Drive, Fairfax, Virginia 22030

Ph: 571-373-9991

Response to Reviewer 2

Thank you very much for reviewing our manuscript. We greatly appreciate you for the complimentary comments and suggestion. The following is our response to the question:

Comment: “The conclusions are very general. Every review part must have the most important conclusion including concrete example or more general trend or direction in gas sensing. Authors must show the most suitable GaN based sensors in one or many cases. It is not easy to follow/understand the author’s main message in conclusions.”

Response: Thanks for pointing this out. In the revised manuscript, we have added the following sentences in conclusion section summarizing the specific findings from our reviews, (Page 18, Lines 459-463) àBased on the analysis of sensing characteristics and the proposed metric, it was found that InGaN/GaN NW sensor shows superior overall sensing performance for H2 gas sensing. Also, GaN/(TiO2–Pt) and GaN/TiO2 NWNC sensors are highly suitable for ethanol and TNT sensing, respectively. Moreover, metal-oxide coated GaN NWs exhibit reliable sensing performance toward various oxidizing gases including NO2 and SO2.

Round 2

Reviewer 1 Report

I feel satisfied with the answers given to my queries and the changes made to the manuscript